# Self-assembled Möbius strips with controlled helicity

Guanghui Ouyang [1,2,6], Lukang Ji[1,3,6], Yuqian Jiang[4], Frank Würthner [2✉] & Minghua Liu [1,5✉]

Different from molecular level topology, the development of supramolecular topology has been limited due to a lack of reliable synthetic methods. Here we describe a supramolecular strategy of accessing Möbius strip, a fascinating topological object featured with only a single edge and single side. Through bending and cyclization of twisted nanofibers self-assembled from chiral glutamate amphiphiles, supramolecular nano-toroids with various twist numbers were obtained. Electron microscopic techniques could clearly identify the formation of Möbius strips when twist numbers on the toroidal fibers are odd ones. Spectroscopic and morphological analysis indicates that the helicity of the Möbius strips and nano-toroids stems from the molecular chirality of glutamate molecules. Therefore, *M*- and *P*-helical Möbius strips could be formed from L- and D-amphiphiles, respectively. Our experimental results and theoretical simulations may advance the prospect of creating chiral topologically complex structures via supramolecular approach.

[1] Beijing National Laboratory of Molecular Sciences, CAS Key Laboratory of Colloid, Interface and Thermodynamics, Institute of Chemistry, Chinese Academy of Sciences, Beijing 100190, China. [2] Institut für Organische Chemie and Center for Nanosystems Chemistry, Universität Würzburg, Am Hubland, Würzburg 97074, Germany. [3] College of Chemistry and Materials Science, Hebei Normal University, Shijiazhuang 050024, China. [4] Key Laboratory of Nanosystem and Hierarchical Fabrication, Chinese Academy of Sciences, National Center for Nanoscience and Technology, Beijing 100190, China. [5] University of Chinese Academy of Sciences, Beijing 100049, China. [6] These authors contributed equally: Guanghui Ouyang, Lukang Ji. ✉email: wuerthner@uni-wuerzburg.de; liumh@iccas.ac.cn

Topology as a mathematical concept[1] to describe three-dimensional objects is of importance for multiple scientific disciplines including biology[2], physics[3], chemistry[4], and architecture[5]. Synthesizing molecules or nanostructures with specific topologies is an attractive challenge for chemists due to their special properties and functions, which may be applied to create functional materials and molecular devices[6,7]. While several kinds of important topologically complex molecular structures such as rotaxane[8], knot[9], and catenane[10], have been fabricated via organic synthesis, their supramolecular analogous nanostructures through self-assembly approach are still considered as rather scarce[11,12]. Among the magical supramolecular topological objects, chiral toroidal structures and especially Möbius strips are of particular interest not only for satisfying curiosity of people but also for understanding myriad chiral toroidal architectures in nature such as circular DNA[13] and proteins[14]. Although several self-assembled circular structures were successfully constructed[15,16], toroidal nanostructures with controlled chirality characteristics have been rarely reported. As a special topological circular structure, a physical Möbius strip can be prepared via half-twisting a band any odd number of times and then fixing the ends together, leading to a topology that has only a single edge and a single side[17]. The first singly twisted Möbius molecule was not prepared until 2003 by Herges and co-workers[18], after which the main strategy for fabricating Möbius strips is total synthesis and several successful examples have been reported[19,20].

Supramolecular chemistry has proven its capabilities for the creation of interesting structures from nano to microscopic levels[21–29]. Although some pioneering reports described Möbius strips from (bio-)macromolecules[12,30] and a NbSe₃ inorganic conductor[31], utilizing self-assembly approach, supramolecular Möbius strips with controlled helicity have thus far not been realized to the best of our knowledge. Here, we report a supramolecular method for the construction of Möbius strips and chiral nanotoroids with controlled helicity. A series of azobenzene-based glutamate compounds with varied alkyl chains has been designed (Fig. 1a and Supplementary Fig. 1). Through a heating-cooling process, these amphiphile compounds were capable of forming twisted nanofibers under the assistance of multiple non-covalent interactions. It was found that pH value variations drove the bending of these twisted fibers due to cooperative adjustment of azobenzene stacking and amide hydrogen bonding. Further lowering amphiphile concentration efficiently shortened the length of bent fibers, which promoted the intrafiber end-to-end cyclization, leading to the formation of chiral toroidal nanofibers in which Möbius strips could be directly visualized by SEM technique. Remarkably, the helicity of the Möbius strips and nanotoroids could be well controlled by using enantiomerically pure amphiphiles. As a result, L-glutamate compounds lead to the formation of M-helical Möbius strips while D-glutamate compounds afford the P-helical ones.

## Results

**Proposed cyclization strategy and assembly protocols**. The proposed strategy for self-assembled Möbius strips and chiral toroids is illustrated in Fig. 1. As a continuation of our ongoing endeavor in developing self-assembled chiral structures from supramolecular gels[32], we hypothesized that glutamic acid amphiphiles functionalized with aromatic moieties (Fig. 1a) could form chiral twisted nanostructures derived from bilayer stacking (Fig. 1b). Owing to the pH responsiveness and stepwise ionization process of the glutamic acid headgroups of these amphiphiles, the curvature of the bilayer and nano-twisted structures formed by self-assembly of these molecules might be tunable by changing the pH value (Fig. 1c). Further elongation of bent twisted fibers might lead to cyclization through intrafiber end-to-end fusion. In this process, the formation of nanotoroids can be promoted through both structural engineering of glutamate molecules and optimization of self-assembly protocols. When the twist numbers on toroid fibers are odd ones, this should prevail in the form of multiple-twisted Möbius strip (Fig. 1d, upper images).

The synthetic procedures of glutamic acid amphiphiles are briefly summarized in the supporting information (Supplementary Fig. 21). In general, these well-designed molecules are composed of three main parts: a chiral glutamic acid head, a planar aromatic azobenzene, and an alkyl tail. The motivation of such molecular design is the following: (1) Though changing the length of alkyl chain, the hydrophobic/hydrophilic balance of the amphiphiles can be effectively adjusted, which significantly

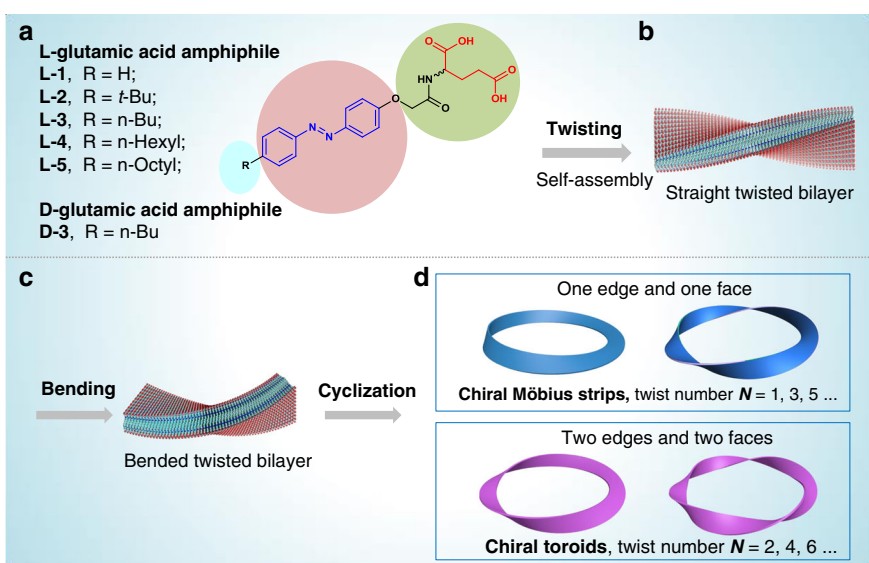

**Fig. 1 Schematic representation of forming supramolecular Möbius strips and chiral nanotoroids through amphiphile self-assembly. a** Molecular structures of L- and D-glutamic acid-derived amphiphiles L-1–L-5 and D-3, respectively. **b** Amphiphiles first self-assemble into straight twisted bilayers, **c** which then bends into bent twisted bilayers. **d** Cyclization of bent fiber leads to the formation of chiral toroidal fibers. When the twist numbers on toroidal fibers are odd ones, chiral Möbius strips can be identified by electron microscopic techniques.

influences their self-assembly capability and resultant nanostructures in water. (2) The azobenzene moiety can promote the self-assembly process by intermolecular π–π interactions. Besides, the introduction of an aromatic moiety is helpful for the analysis of the self-assembly process through spectroscopic techniques such as UV–vis absorption and circular dichroism spectroscopy. (3) The glutamic acid head can provide chirality information for the twisted self-assemblies. Most importantly, the ionization process of glutamic acid can be varied at different pH values, influencing the molecular configurations and components of the amphiphiles as revealed by DFT computations and the Henderson–Hasselbalch equation (pH = pKa + log([base]/[acid])), respectively, which would contribute to drive the bending and further cyclization of twisted nanofibers.

The self-assembly protocol of these amphiphiles was performed using typical heating-cooling procedures. Our results demonstrate that their self-assembly capability in aqueous media varies greatly depending on the terminal alkyl groups as expected. L-1 and L-2, which contain a hydrogen and a *t*-butyl group at the end of the azobenzene unit, respectively, failed to form hydrogels at various pH values. Instead, only precipitates could be found (Supplementary Fig. 1a, b). Amphiphiles L-3, L-4, and L-5, which contained four, six, and eight carbon chains on the terminal, respectively, could form hydrogels as evidenced by vial-inversion trials (inserted images, Supplementary Fig. 1c, d, f), indicating that a linear alkyl chain is critical for successful self-assembly. When lowering the pH value to 2.0, L-3, L-4, and L-5 could still form stable hydrogels ([amphiphile] = 3.0 mg/mL), but the nanostructures were distinctly different. The SEM images of the L-3 and L-4 hydrogels revealed that these molecules could form bent nanotwists at pH 2.0. As expected, some of them bent into uncommon toroidal structures (Supplementary Fig. 1c–e). However, the nanotwists in L-5 hydrogel were almost straight and no nanotoroid was observed via SEM images (Supplementary Fig. 1f). These results suggest that the alkyl chain at the end of the molecules has a dramatic influence on self-assembly ability. Most importantly, the length of the alkyl chain was crucial for the bending of nanotwists under pH variation, especially for the formation of nanotoroids.

Although there were more nanotoroids in the hydrogel formed by L-4 compared to L-3, the twist directions of L-4 assemblies were not distinguishable (Supplementary Fig. 1g, h). As a result, L-3 was chosen as the candidate for the following in-depth studies. The hydrogel of L-3 was formed among wide pH values ranging from 2.0 to 7.0, as shown in Supplementary Fig. 2. Upon gelation, the compound L-3 self-assembled into left-handed nanotwist structures observed by SEM (Fig. 2a, Supplementary Fig. 3), which indicate that chirality was successfully transferred from molecular to supramolecular level. These nanotwist structures were essentially long and straight at a higher pH value (pH ≥ 3, Supplementary Fig. 3b–f). However, they had a tendency to bend when lowering pH value as expected (Fig. 2b, Supplementary Fig. 3a) and uncommon chiral toroidal structures appeared when the pH value reached 2.0, although they structurally remained fibers (Fig. 2c, d).

The two key requirements for the successful formation of self-assembled chiral toroids are the bending and cyclization processes of twisted fibers as illustrated in Fig. 1. The bending of twisted fibers has been realized by adjusting the pH value. Due to the absence of suitable template like those utilized in the synthesis of toroidal molecules, the cyclization of bent fibers is much more challenging. Theoretically, the two terminals of a gradually elongating bent fiber will meet if the curved precursor remains in a plane. However, as the assemblies are growing in three-dimensional aqueous media, their elongation direction is generally random. A reasonable method is lowering the

concentration of amphiphiles and therefore shortening the length and reducing the amount of twisted fibers, which will enhance the probability of intrafiber end-to-end cyclization by suppressing interfiber fusion. When keeping the pH value fixed at 2.0, L-3 could still form hydrogels when lowering the concentration from 6.80 to 1.13 mM, giving mainly long fiber structures as demonstrated by SEM images (Supplementary Fig. 5a–d). Further lowering the concentration of L-3 from 0.45 to 0.05 mM led to the formation of floccule structures, which were composed of nanofibers with shortened length as revealed by SEM images. Most importantly, the majority of these short fibers became bent and the amount of nanotoroids also increased significantly (Supplementary Figs. 5e–i, 6, 7). To evaluate the population of chiral toroidal structures among the whole nanofibers, we statistically analyzed the morphological parameters of toroidal fibers. As shown in Supplementary Fig. 8, the ratio of circumference of chiral toroidal structures to the length of the whole nanofibers including toroids and uncyclized fibers is calculated to be about 50%, which is obviously larger than that of L-3 assemblies at higher concentration (Supplementary Fig. 5). Through the combination of these two simple methods, lowering pH value and reducing amphiphile concentration, the number of chiral nanotoroids among uncyclized fibers is effectively improved.

**Statistical analysis and identification of Möbius strips**. We further statistically analyzed the distribution of twist number, diameter, and fiber width among 95 nanotoroids observed through scanning electron microscopy images of L-3 amphiphile. As shown in Supplementary Figs. 9–11, when the twist numbers were small (≤3), we can clearly recognize the exact twist numbers and helicity of these chiral toroids. Singly twisted toroids are the dominant species among those with small twist numbers as shown in Fig. 2i and Supplementary Fig. 9. Although the helical chirality of the other chiral toroids could be clearly unmasked by SEM technique, the exact twist numbers remain impossible to count due to overcrowded twists in the small-sized ring fiber (Supplementary Fig. 12). Diameter and fiber width analysis of these 95 chiral nanotoroids indicated that the diameters of majority toroids are among 500–2500 nm and fiber width is among 60–230 nm. According to the diameters, the average circumference of these circular fibers is in the range 1.6–7.8 μm, which is obviously shorter than the length of long fibers at high concentration (Supplementary Fig. 5). These results implied that the length of bent fibers has a significant influence on the success of cyclization, which indirectly supported that the formation of chiral circular structures can be promoted by lowering the amphiphile concentration to reduce the amount and shorten the length of bent fibers.

SEM images indicated that all of these toroids are chiral and those with odd twist number can be clearly recognized as Möbius strips by electron microscopic technologies. As shown in Fig. 3a, when a reptile begins its crawl from point S1 on the outer face of a toroidal fiber, it will go across a twist (green dotted arrow) and then arrives at point S2 on the inner face. After it walks across the whole inner face (green solid arrows), it will arrive at point S3 and then go across the twist (yellow dotted arrow) and arrive at point S4. After it walks across the whole outer face (yellow solid arrows), it will finally return to the starting point S1. These walk paths indicated that the nanotoroid showed in Fig. 3a is actually a Möbius strip. Same phenomenon can also be found for nanotoroids showed in Fig. 3b and Supplementary Fig. 9. These nanotoroids are found in the assembly structures of L-3 and the twist helical direction showed *M* chirality. Repetition tests demonstrated that the Möbius strips and chiral nanotoroids

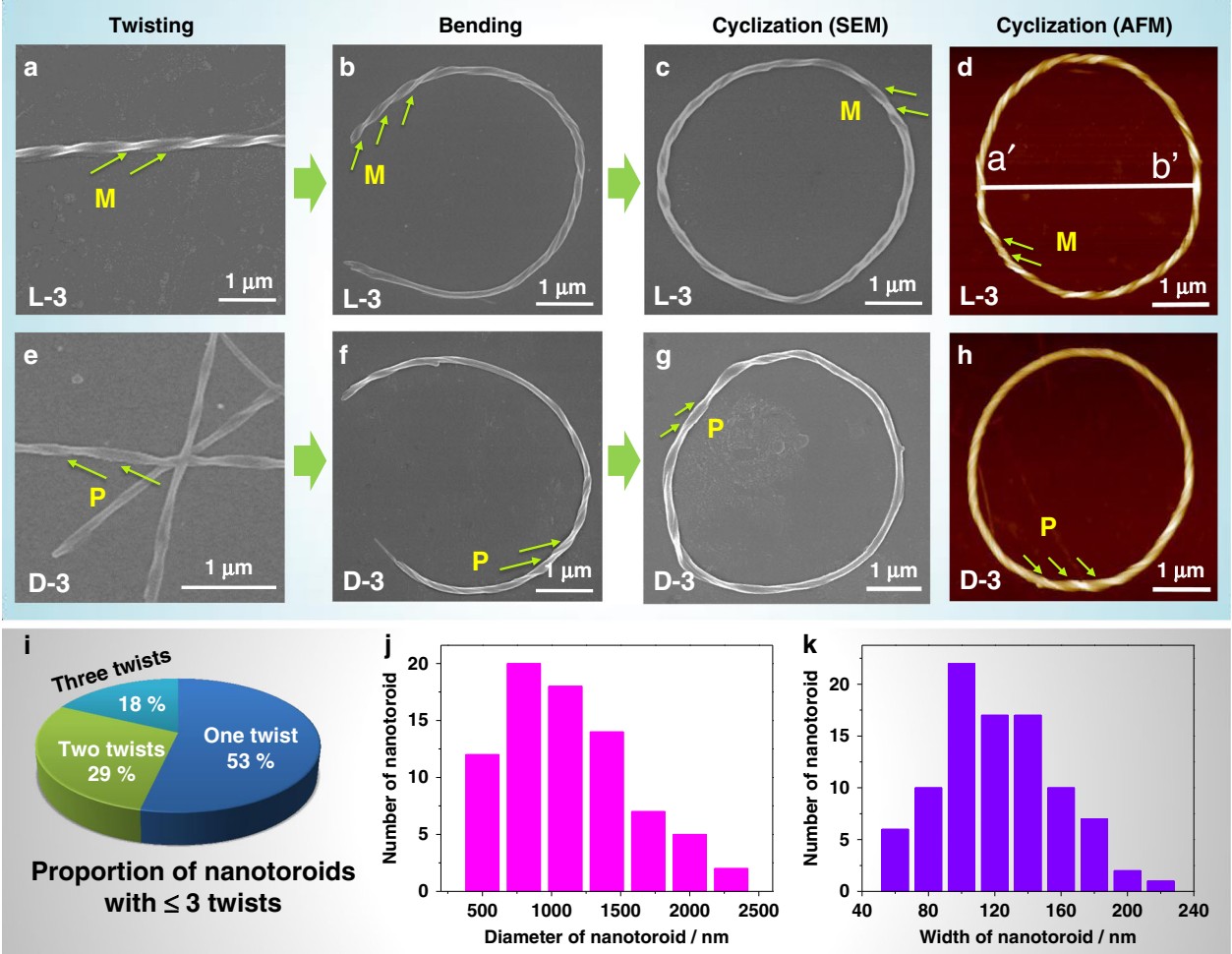

**Fig. 2 Bending and cyclization processes of twisted nanofibers revealed by electron microscopic images and statistical analysis of chiral toroidal fibers.** SEM and AFM images of L-3 self-assemblies showing *M* chirality. **a** Linear twisted fiber. **b** Semicircle shaped twisted fiber. **c**, **d** Chiral toroidal fibers, the diameter (a′–b′) of toroid showed in (**d**) is about 4.0 μm. SEM and AFM images of D-3 self-assemblies showing *P* chirality. **e** Linear twisted fiber. **f** Semicircle shaped twisted fiber. **g**, **h** Chiral toroidal fibers. [L-3] = [D-3] = 0.11 mM, pH = 3 for (**a**) and (**e**), pH = 2 for (**b**–**d**) and (**f**–**h**). Notes: *M* means left helicity and *P* means right helicity. **i**–**k** Statistical analysis of morphological properties of nanotoroids observed from SEM images of L-3 self-assemblies (total number of nanotoroids analyzed, 95). **i** Proportion of nanotoroids with ≤3 twists on the toroidal fibers, the number of toroids with one, two, and three twists are 16, 8, and 5, respectively. **j** The frequency distribution histogram is plotted against the diameter of nanotoroids. **k** The frequency distribution histogram is plotted against the width of nanotoroids.

could be found at various batches of assemblies at pH 2.0, showing that the formation of Möbius strip and chiral nanotoroids in this system was reliable and not accidental (Supplementary Figs. 9–14).

Möbius strips are intrinsically chiral by virtue of their topology. However, effectively controlling the chirality of a Möbius strip remains challenging partly due to the torsion of twist structure[33]. Möbius strip molecules or structures reported in previous literatures all appeared in the form of racemates or mixtures with both right-handed and left-handed chirality[12,18,19,30]. Owing to effective chirality transfer, all the chiral nanofibers from L-3 assemblies showed *M*-helicity, which implied the origin of supramolecular chirality in this system. We therefore further synthesized the corresponding enantiomer D-3 (Fig. 1a) and investigated its self-assembly in the same way as that for L-enantiomer. As expected, right-handed Möbius strips (Fig. 3c) and nanotoroids (Fig. 2g, h, Supplementary Fig. 14) were observed via AFM and SEM microscopic techniques. These results confirmed that L-3 led to the formation of left-handed Möbius strips, while D-3 led to the right-handed ones. The supramolecular chirality of these nanoassemblies was further

confirmed via the circular dichroism (CD) spectra. The amphiphile self-assemblies were composed of chiral toroids and uncyclized fibers (Fig. 2, Supplementary Figs. 8, 12). Their helical chirality can be clearly proved to be the same by SEM and AFM images. Therefore, the Cotton effects revealed by CD spectra could be used to identify the chiral contribution of both chiral toroids (including chiral Mobius structures) and uncyclized fibers. As shown in Fig. 4e, L-3 and D-3 self-assemblies gave mirror-imaged CD signals with a bisignate Cotton effect with a crossover at about 325 nm, which almost corresponds to the absorption maximum of azobenzene. When mixing L-3 and D-3 in a 1:1 molar ratio, the obtained nanotoroids showed no helical chirality according to SEM observation (Fig. 3d, Supplementary Fig. 15), which was also supported by the CD spectrum showing silent signal (Fig. 4e, green dotted line). These results demonstrated that the helicity of the Möbius strip stemmed from the original molecular chirality of the amphiphiles. Thus, a supramolecular method for the preparation of Möbius strips and nanotoroids with controlled helicity has been successfully established through the self-assembly of well-designed amphiphile molecules.

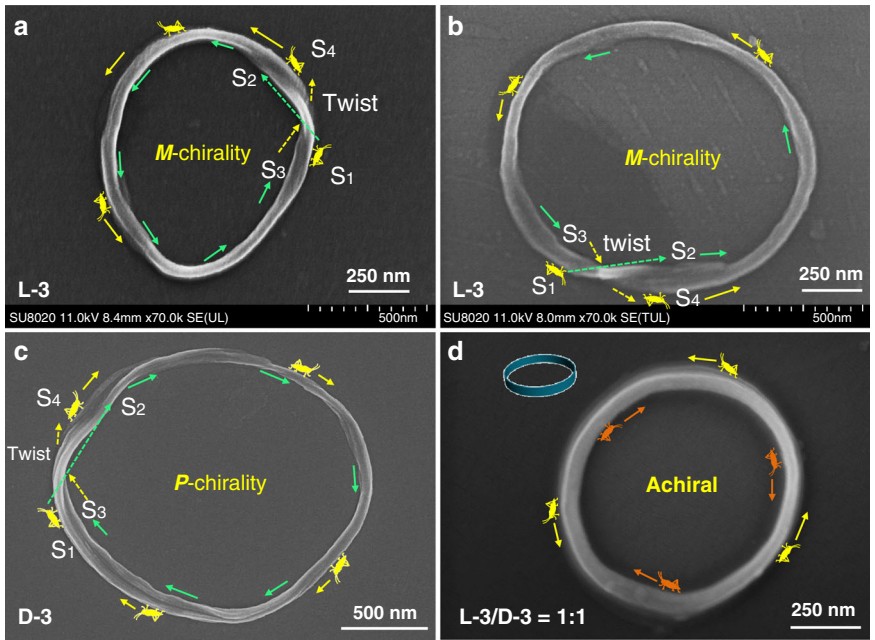

**Fig. 3 Identification of chiral Möbius strips. a**, **b** SEM images of L-3 assemblies showed Möbius strips with a single twist on the fiber. **c** SEM image of D-3 assemblies showed a Möbius strip with a single twist on the fiber. The yellow and green arrows are used to represent the walk direction of a reptile on the toroidal fibers. **d** SEM image of L-3/D-3 (mole ratio 1:1) assemblies showing an achiral nanotoroid. The yellow and orange arrows are used to indicate the walk directions of two reptiles on the outer and inner faces, respectively. The concentration of amphiphile is 0.11 mM, pH = 2. Notes: *M* means left helicity and *P* means right helicity.

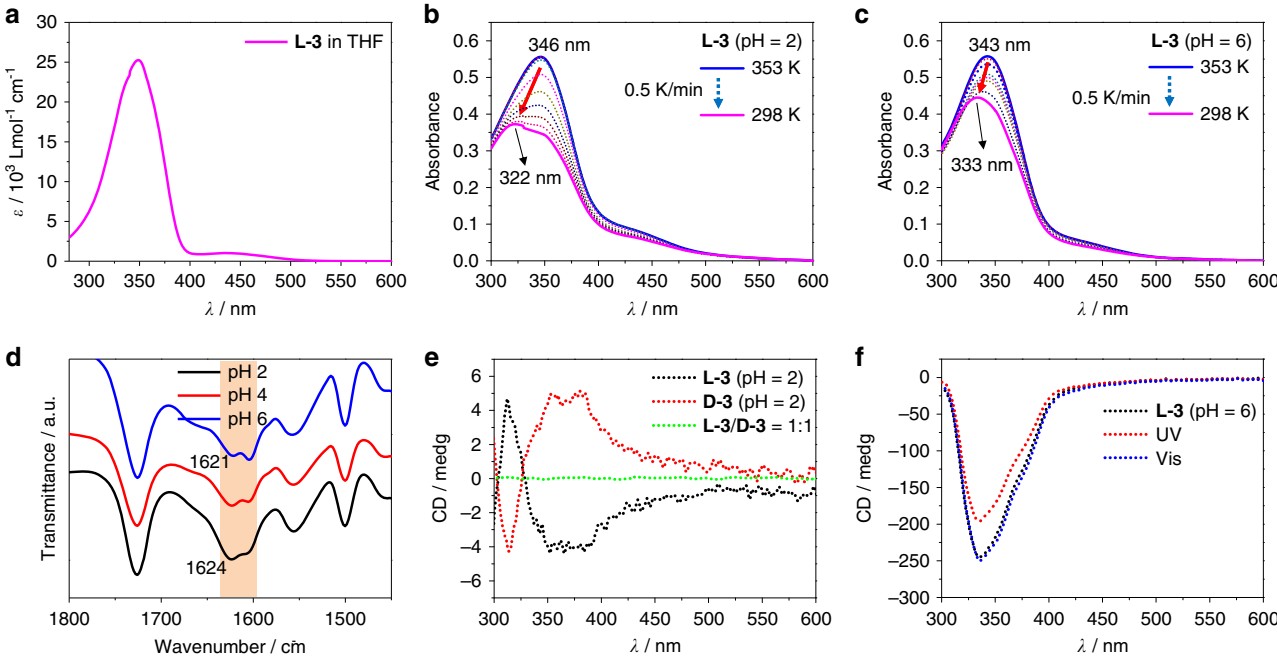

**Fig. 4 Spectroscopic study of amphiphile self-assembly. a** UV–vis spectrum of L-3 amphiphile in THF. [L-3] = 50 µM, 298 K, $\varepsilon_{max} = 2.5 \times 10^4\,M^{-1}\,cm^{-1}$ at 348 nm. **b** Variable-temperature UV–vis spectra of L-3 amphiphile in water. [L-3] = 50 µM, pH = 2. Cooling speed is 0.5 K/min, data interval is 5 K/record. **c** Variable-temperature UV–vis spectra of L-3 amphiphile in water. [L-3] = 50 µM, pH = 6. Cooling speed is 0.5 K/min, data interval is 5 K/record. **d** FT-IR spectra of self-assemblies of L-3 amphiphile in water at different pH values, blue line: pH = 6, red line: pH = 4, black line: pH = 2, [L-3] = 0.11 mM. **e** CD spectra of self-assemblies of L-3 (black dotted line), D-3 (red dotted line) and their 1:1 mixture (green dotted line), [L-3] = [D-3] = 50 µM, pH = 2. **f** CD spectra of self-assemblies of L-3, [L-3] = 50 µM, pH = 6, black dotted line: normal L-3 assemblies, red dotted line: L-3 assemblies after UV 365 nm irradiation for 1 h, blue dotted line: L-3 assemblies after visible light irradiation for 1 h.

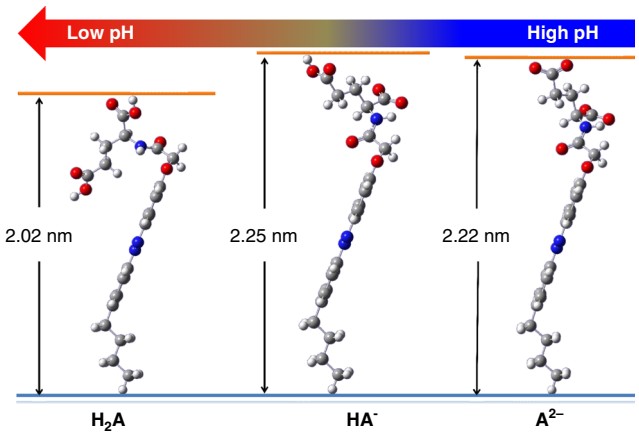

**Fig. 5 DFT computation of L-3 amphiphile at different ionization states.** Gaussian B3LYP/6-31+G* optimization was utilized, the red, blue, dark gray, and light gray balls represent O, N, C, and H atoms, respectively. $H_2A$ represents L-3, while $HA^-$ and $A^{2-}$ are its ionized counterparts.

**Spectroscopic experiments and computational simulation**. To get insight into the self-assembly and cyclization mechanism of these amphiphile molecules, a series of spectroscopic experiments and computational simulation were conducted. Due to the existence of two carboxylic acid groups, the glutamic acid head of L-3 amphiphile should show different ionization levels at varied pH values, which might influence the molecular configuration and therefore assembly mode. Using the symbol $H_2A$ instead of L-3, its ionization equation is as follows:

$$H_2A + H_2O = H_3O^+ + HA^-$$
$$HA^- + H_2O = H_3O^+ + A^{2-} \tag{1}$$

Density functional theory (DFT) calculation using B3LYP/6-31+G* demonstrated that the molecular lengths of $HA^-$ and $A^{2-}$ are similar (2.25 and 2.22 nm, respectively), however, $H_2A$ has a shortened length to about 2.02 nm (Fig. 5). According to previous research[34], the pKa values of glutamic acid are pKa1 = 2.16 and pKa2 = 4.32. Based on Henderson–Hasselbalch equation:

$$pH = pKa + \lg([base]/[acid]) \tag{2}$$

At higher pH values (≥3), the major components of L-3 amphiphile molecules are $HA^-$ and $A^{2-}$, which have almost equal molecular length. When lowering the pH value, the fraction of $H_2A$ increases, and when pH = 2, the major components in the L-3 system are $HA^-$ and $H_2A$. The ratio of $HA^-/H_2A$ is calculated to be 0.69 according to Eq. (2).

Spectroscopic studies were carried out to obtain insights into the aggregation of these amphiphiles. The absorption behavior of L-3 in THF solution was shown in Fig. 4a. The major absorption band at 348 nm was ascribed to the *trans*-azobenzene π–π* transition, while the minor absorption peak at 445 nm was corresponding to the n–π* transition of *cis*-azobenzene[35]. To investigate the aggregation process of L-3 at different pH values, variable-temperature UV–vis spectroscopy was applied. L-3 amphiphile was dispersed in water and heated at 353 K for 30 min, a machine-controlled cooling process at a speed of 0.5 K/min was utilized to record the UV–vis spectra. Results showed that the hypsochromic shift of azobenzene moiety during self-assembly process is more obvious (from 346 to 322 nm, Fig. 4b) when lowering the pH value from 6 to 2, which implies a more compact aggregate structure at lower pH value[36]. The compact aggregate of azobenzene would suppress its photo-isomerization ability, which was proved by the photo-isomerization

experiments. As shown in Fig. 4f, the CD spectra of L-3 assemblies at pH 6 showed photo-responsiveness under UV 365 nm irradiation, while it showed no obvious photo-response when the pH value was lowered to 2. These CD results indicate that the π–π stacking is stronger at lower pH value, which is consistent with the absorption spectra.

XRD pattern showed that the d-spacing values of L-3 self-assemblies at different pH values are between 3.29 and 3.90 nm (Supplementary Fig. 4), which were larger than one but less than twice the molecular length of L-3, indicating the formation of bilayer structures. The layer distance 3.66, 1.85, 1.21, 0.91, 0.73, and 0.61 nm of L-3 at pH 2.0, indicates a well-defined lamellar structure with a d-spacing of 3.66 nm. CD spectra of L-3 assemblies at pH = 6 showed a normal negative Cotton effect (Fig. 4f, black dotted line), while a bisignate Cotton effect was observed for L-3 assemblies at pH = 2 (Fig. 4e). According to exciton chiral theory[37], the negative bisignate Cotton effect showed in Fig. 4e implies a counterclockwise screw sense of the two transition moments of the adjacent azobenzene moieties. Therefore, a reasonable aromatic stacking mechanism based on these data was proposed (Supplementary Fig. 16). At higher pH value, the bilayer structures adopt a parallel stacking mode and further assemble into twisted multi-bilayers and fibers through hierarchical self-assembly (see below). At lower pH values, the major components of amphiphile molecules are $H_2A$ and $HA^-$, the shorter molecular length of $H_2A$ led to a rotation between two adjacent bilayers (Supplementary Fig. 16). As a result, the CD signal showed a bisignate Cotton effect with a crossover at the absorption maximum region of azobenzene. Besides, the rotation of bilayer stacking should also influence the hydrogen bonding of amide groups. This was well supported by FT-IR spectra, which showed that the C=O stretching vibration of amide bond was changed from 1621 to 1624 nm when lowering pH values from 6 to 2, indicating that the hydrogen bonding between amide groups was weakened (Fig. 4d).

Molecular dynamic simulation (MD simulation) was further applied to provide insights for the bending process of bilayer structures. We built a pre-assembled bilayer with planar structure containing 900 L-3 molecules with 45*10*2 array to study the $HA^-/H_2A$ aggregate via MD simulation, as shown in Fig. 6a (for easy calculation, we set the ratio of $HA^-/H_2A = 1/1$, the magenta molecules are $H_2A$, while cyan molecules are $HA^-$). After MD simulation, the equilibrium configuration of the $HA^-/H_2A$ bilayer was extracted as shown in Fig. 6b, c. The bilayer here was bent and twisted at the same time. The headgroups of $HA^-$ molecules point to the outside of the curve, while those of $H_2A$ molecules point to the inside. These simulation results explain how the linear twist at higher pH values turns into a twisted one when lowering the pH to 2.0. The singly twisted Möbius strip can be described mathematically by the following equations[38], where $t$ represents rotation angle, $R$ represents the radius of Möbius strip, $s$ represents the half-width of band.

$$x = \left[R + s * \cos\left(\tfrac{1}{2}t\right)\right]\cos t$$
$$y = \left[R + s * \cos\left(\tfrac{1}{2}t\right)\right]\sin t$$
$$z = s * \sin\left(\tfrac{1}{2}t\right)$$
$$\text{for } s \in [-w, w], t \in [0, 2\pi] \tag{3}$$

According to our MD computation data, the width of the MD twist is about $2w = 7.7$ nm, and the curvature radius ($R$) of the MD twist is about 34.4 nm (Fig. 6b, c). The calculated ratio of $R/w$ is about 8.9, which was used to obtain the standard Möbius strip images based on Eq. (3) and equations in Supplementary Fig. 17. As shown in Fig. 6d–f, the upper images are theoretical singly, triply, and quintuple twisted Möbius strips, respectively. The

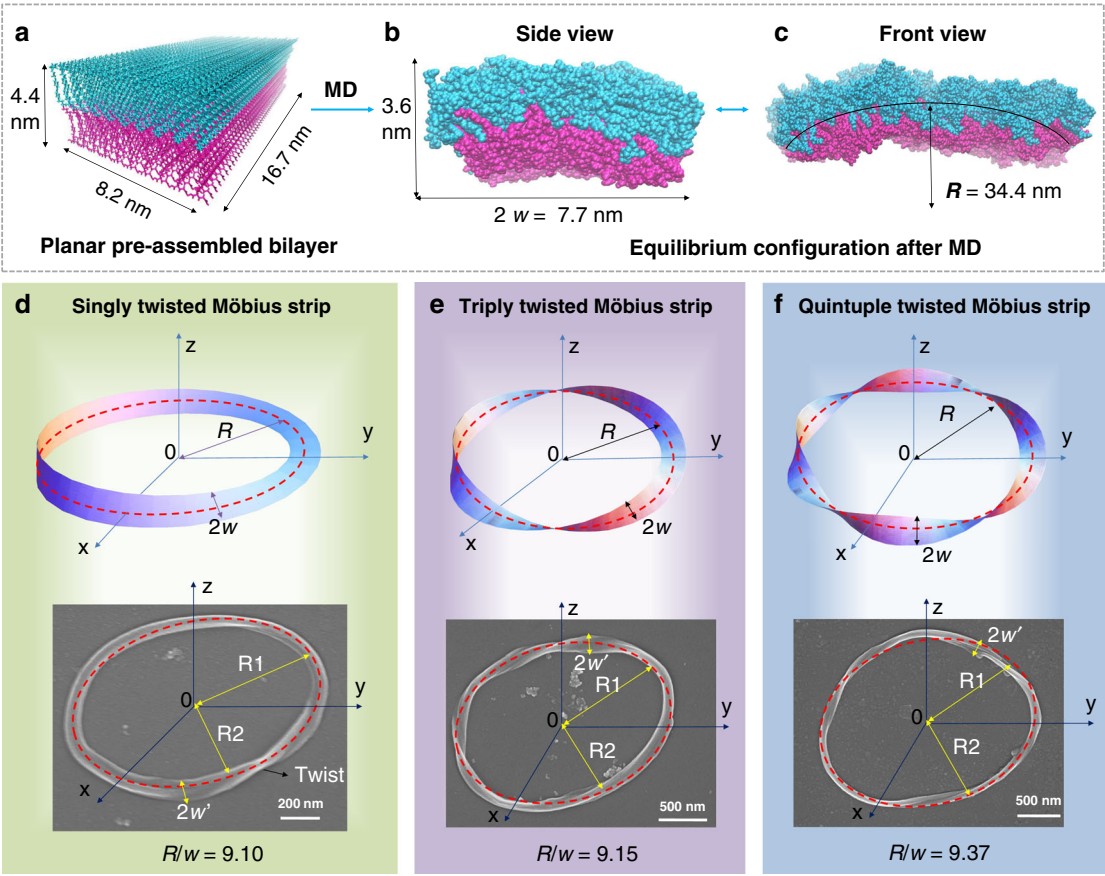

**Fig. 6 Molecular dynamic (MD) simulation and mathematical analysis. a** Planar pre-assembled bilayer structure containing 900 molecules with 45*10*2 array, $H_2A$, and $HA^-$ molecules were colored in magenta and cyan, respectively. **b** The equilibrium configuration for $H_2A/HA^-$ after MD simulation, side view, the length of the bilayer is about 3.6 nm, the width of the twist is about 7.7 nm. **c** The equilibrium configuration after MD simulation, front view, the curvature radius of the twist is about 34.4 nm. **d–f** Theoretical Möbius strips (upper images) image based on parameters from MD computation and experimental Möbius strips (bottom images) observed from SEM images with different twist numbers. **d** Singly twisted Möbius strip. **e** Triply twisted Möbius strip. **f** Quintuple twisted Möbius strip. Notes: R represents the radius of Möbius strip, R1 and R2 represent the long radius and short radius of elliptic Möbius strip, w and w' represent the half-width of Möbius strip band. $H_2A$ = L-3.

topological parameters of experimental observed Möbius strips were analyzed to compare with theoretical ones. As shown in Fig. 6d (bottom image), we measured the singly twisted Möbius strip observed by SEM, its topology parameters are as follows: the average width of strip is about $2w = 110$ nm, and the average radius R is 500 nm, so the experiment ratio $R/w$ is 9.1, which is consistent with the calculated one from MD. For triply and quintuple twisted Möbius strips observed by SEM, same analytical methods were adopted, and the corresponding experimental ratios of $R/w$ were 9.15 and 9.37, respectively (bottom images of Fig. 6e, f and Supplementary Fig. 17), which were also almost consistent with the calculated MD $R/w$ ratio. These results help us to understand the mathematical parameters and the formation process of Möbius strip observed in this report and further supported the identification of Möbius strips.

## Discussion

Based on the above experimental data and computational results, a possible cyclization mechanism for the formation of chiral Möbius strips and nanotoroids was proposed. As shown in Fig. 7b, the bent twisted fibers generally have two major elongation pathways. When the two ends of the bent fibers are not in a plane, further elongation leads to the formation of uncyclized twisted fibers. When two ends are in a plane (this probability is higher for shorter fibers due to self-standing of twisted

structures), the elongation would produce chiral toroidal fibers, whose helicity can be visualized by electron microscopic images. Especially, when the twist numbers on these nanotoroids can be clearly recognized and are odd ones (Supplementary Figs. 9, 11), the identification of chiral Möbius strips are successfully achieved. During the cyclization process of bent twisted fibers, other unique nanostructures including catenane-like nanostructures (Supplementary Fig. 18b, c) and uncyclized concentric disc structures (Supplementary Fig. 18d) were also observed in the self-assemblies of amphiphile molecules. These structures are most-likely the by-products due to failure of cyclization, which indirectly implied that the chiral nanotoroids and Möbius strips were formed via cyclization of bent nanotwists.

In summary, multiple twisted macroscopic Möbius strips and nanotoroids were obtained through self-assembly strategy. The helicity of the Möbius strips are fully controlled by the molecular chirality of the starting organic molecules, which may become a general strategy for other types of Möbius strips toward chiral topological supramolecular functional materials. As a proof of principle, we have used the Möbius strips as a template skeleton for preparing luminescent chiral Möbius strips by doping various organic dyes (Supplementary Figs. 19–20). The operationally simple method provides a powerful way to access Möbius strips and chiral nanotoroids with multiple twists and specific chirality. We believe that the experimental and theoretical simulation results reported here will advance the prospect of supramolecular

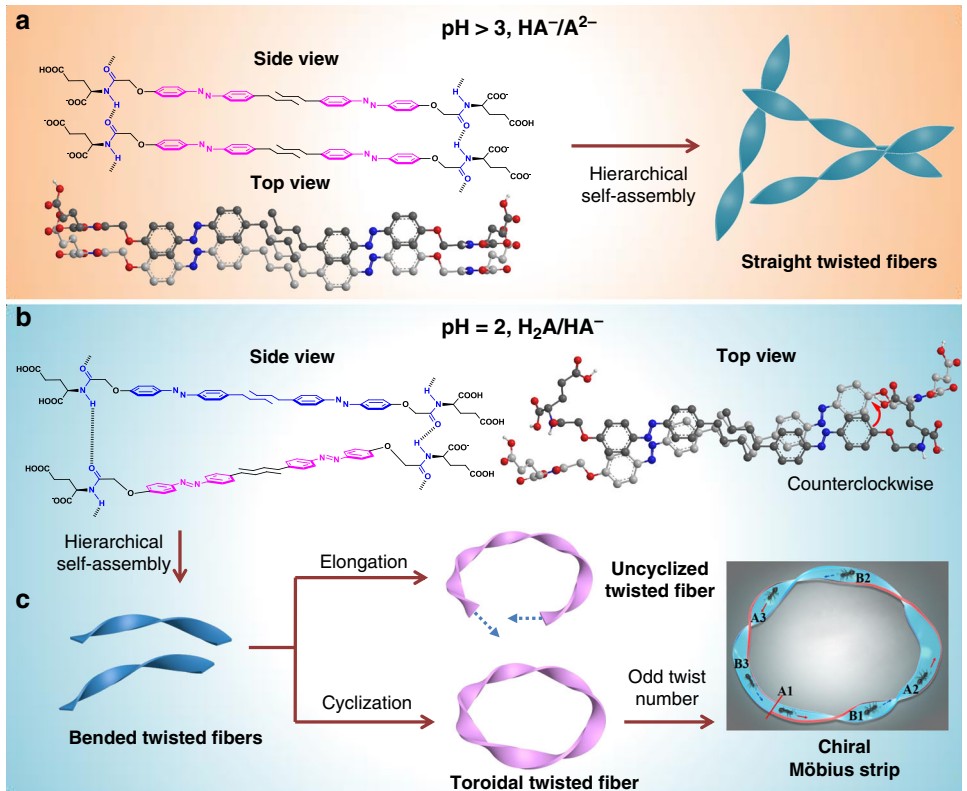

**Fig. 7 Schematic illustration of proposed molecular stacking mode, self-assembly, and cyclization process. a** When pH >3, the major components HA$^-$ and A$^{2-}$ of L-3 amphiphile molecules form a bilayer structure mainly through azobenzene π-π stacking and hydrogen bonding between amide groups, the tiny molecular length difference of HA$^-$ and A$^{2-}$ lead to the formation of straight twisted fibers. **b** When pH = 2, the major components H$_2$A and HA$^-$ of L-3 amphiphile molecules form a counterclockwise bilayer stacking due to the obvious molecular length difference of H$_2$A and HA$^-$. **c** Illustration of two possible elongation pathways for bent twisted fibers. When the two fiber terminals are not in a plane, further elongation leads to the formation of uncyclized twisted fibers, otherwise, further elongation leads to the cyclization of twisted fibers and formation of chiral nanotoroids and Möbius strips. The cartoon figure at the bottom right corner represents a quintuple twisted Möbius strip, if an ant starts from side A1 and proceeds sequentially through B1 backside, A2, B2 backside, A3 and B3 backside (red solid arrows), it will finally arrive at the backside of the starting side A1. Upon continuing for a second cycle through A1 backside, B1, A2 backside, B2, A3 backside and B3, the ant will finally arrive back at the starting point.

technology for creating topologically complex structures with controlled helicity.

## Methods

**Materials**. All the starting materials were purchased from TCI company and used as received without further purification. Milli-Q water (18.2 MΩ·cm) was used in the assembly tests. The synthetic route of amphiphiles was listed in the Scheme S1 of the Supporting Information.

**Self-assembly protocol**. The formation of the Möbius strips and chiral nano-toroids went through a heating-cooling process. Generally, certain amounts of amphiphiles was added into Milli-Q water, and diluted hydrochloric acid was used to adjust the pH value to 2.0. Afterward, the mixture was heated to 90 °C for 30 min and a transparent solution was obtained. Then the solution was cooled to room temperature (25 °C) to give an orange gel (higher amphiphile concentration) or dispersion (lower amphiphile concentration) depending on the amounts of amphiphiles used. The Möbius strips could be observed via electron microscopy techniques.

**Scanning electron microscopy (SEM), atomic force microscopy (AFM), and transmission electron microscopy (TEM)**. Samples were cast onto single-crystal silica plates, the solvent was evaporated under ambient conditions, and then vacuum-dried. The sample surface was coated with a thin layer of Pt to increase the contrast. SEM images were recorded on a Hitachi S-4800 FE-SEM instrument and a Hitachi SU-8020 instrument with an accelerating voltage of 10 kV and 11 kV, respectively. AFM images were obtained on a Dimension FastScan (Bruker), using ScanAsyst mode under ambient condition. Fastscan B probes were used for the scan, and samples were prepared by dropping samples onto a mica sheet. TEM images were obtained on a JEM-1011 electron microscope at an accelerating

voltage of 100 kV. The TEM samples were prepared by casting a small amount of sample on carbon-coated copper grids (300 mesh) and dried under strong vacuum.

**X-ray diffraction (XRD) measurements**. XRD analysis was performed on a Rigaku D/Max-2500 X-ray diffractometer (Japan) with Cu Kα radiation (λ = 1.5406 Å), which was operated at a voltage of 40 kV and a current of 200 mA. Samples were cast on glass substrates and vacuum-dried for XRD measurements.

**Ultraviolet-visible spectroscopy experiments and circular dichroism (CD) measurements**. UV–Vis spectra were recorded in quartz cuvettes on JASCO UV-550 and UV-660 spectrometers. CD spectra were recorded in quartz cuvettes on a JASCO J-810 spectrophotometer.

**Fourier transform infrared (FT-IR)**. FT-IR spectra were recorded on a Bruker Tensor 27 FT-IR spectrometer at room temperature. The KBr pellets made from the vacuum-dried samples were used for FT-IR spectra measurements.

**$^1$H NMR spectra and mass spectra**. $^1$H NMR (400 MHz) spectra were recorded on a Bruker Avance 400 spectrometer with TMS as internal standard at 298 K. Mass spectral data were obtained by using a BIFLEIII matrix-assisted laser deso-rption/ionization time of fight mass spectrometry (MALDI-TOF MS) instrument.

**Molecular dynamics (MD) simulation method**. The pre-assembled aggregates of bilayers are solvated in H$_2$O boxes. Energy minimization was performed for H$_2$A/HA$^-$ bilayer solution system before MD simulation. With a time-step of 2 fs at 298 K, a 500 ps MD simulation within NVT ensemble was first carried out to make the system fully relaxed. Then a 3 ns MD simulation with NPT ensemble was performed based on NVT equilibrium configuration. Berendsen thermostat with a time-step of 1 fs was employed to regulate the temperature at 298 K during NPT

simulation. All the calculations were performed in GROMACS-4.6.7[39] by using general Amber force-field (GAFF)[40,41].

## Data availability
All relevant data are available from the authors. Supplementary Information is available in the online version of the paper. Source data are provided with this paper.

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

## Acknowledgements
This work was supported by financial support from the Chinese Academy of Sciences (QYZDJSSW-SLH044), National Natural Science Foundation of China (21890734, 21890730, 21861132002, 21971247), and Youth Innovation Promotion Association of CAS (2019036).

## Author contributions
M.H.L. supervised the project. M.H.L. and F.W. conceived the work. G.H.O. and L.K.J. performed the experiments and analyzed the data, Y.Q.J. made the computational simulation. G.H.O. and M.H.L. analyzed the mathematical simulation. G.H.O., L.K.J., F.W., and M.H.L. co-wrote the manuscript.

## Competing interests
The authors declare no competing interests.
