## [Peer Review File · Nature Communications]

Reviewer #1 (Remarks to the Author):

This paper describes the formation of cyclic nanofibers by glutamate amphiphiles functionalized with azobenzene. In particular, the authors focus on the twisting of the circular structures and claim that they are Moebius strips on the assumption that they twist an odd number of times. The formation of the microscale circular structure is interesting (but not new, as reported for polymers, small molecules, biomolecules, and carbon nanotubes), but the main point of this paper, especially the identification of the Moebius structure, remains a major question. Although the authors claim generality of the present example many times, the given example is a highly specific self-assembly phenomenon, and the discussion is not in depth with many ambiguous points in principle (given below). The authors presented good figures, carried out complex calculation and intricate analysis, but failed to show the principal mechanism. Therefore, the referee does not agree the publication of this paper in Nature Communications. In my opinion, Langmuir or ACS is the best journal to submit this paper.

The most important issues are listed below.

- Moebius structures might be formed in many linear fibers and the authors mention the term “yield” many times in the manuscript, but they have not quantitatively analyzed the population of the structures.
- The authors describe the preparation of the Moebius structure as a simple strategy. But, which section of the preparation is strategic? What are the motivations of such molecular design and preparation protocol to obtain Moebius structure? Do the circular structures form accidentally? Which factor drives the cyclization of nanofibers?
- The identification of the Moebius structure must be difficult for readers because the ribbon-like structures are ambiguous, and therefore the definition of their face is also ambiguous. For example, are the structures shown in Fig. 1 g, 1k Moebius?
- Thickness of the fibers forming Moebius structures is ambiguous. Although it is clear from XRD that the internal structure is a bilayer motif, the periodicity of about 4 nm means a multilamellar structure. Therefore, the results of MD simulation with the single bilayer structure are not reasonable.
- The essential mechanism for the odd-number times of twisting is not described.
- The authors analyze the specific chirality of Moebius strips using CD spectroscopy. However, this analysis is questionable because the CD spectra contain large contribution of uncyclized fibers. Therefore, the CD-spectra cannot explain the specific chiral contribution of the Moebius structures.
- There are many additional errors in the manuscript, for example, P5, (Figs. 1g, 1h, 1j, 1l, S3a) is (Figs. 1g, 1k, 1j, 1l, S3a)? P7, (Fig. S5) is (Fig. S4)?

P7, (Fig. S6) is (Fig. S4)?

Reviewer #2 (Remarks to the Author):

In my personal opinion, publication of images of self-assembled Moebius strips (and their helicity control) themselves is valuable enough. This is professional work on masterpiece products of self-assembly. I respect this work. I basically recommend publication of this work in Nat. Commun. This work is well done and well considered. However, some revisions are necessary.

1) The contents of this work probably satisfies curiosity of general readers. However, it might not be perfectly satisfactory for specialists in field of molecular assembly. Molecular-structure-based explanation to rationalize molecular interaction in structure-level to mesoscopic morphologies is not so clear. At least, such molecular-interaction-based explanation with chemical formulae had better be presented (proposed) in supporting information.

2) Similarly, description in Introduction had better include more about deep molecular assembly science with citing recent comprehensive papers on self-assembly (see, Bull. Chem. Soc. Jpn. 91, 623-668 (2018), Mol. Sys. Des. Eng. 4, 11-28 (2019).). Current Introduction is rather too much oriented topology and appearance on Moebius strips.

3) Please avoid several easy mistakes. References 35-37 remains at unnatural place. Reference styles are not perfectly unified (comma, colon, etc)

4) Please remove unnecessary grey background from Figure 2j. I do not care about imperfect nature of CD spectra of chiral assemblies. It may happen.

5) Please provide clear scale bars and scale values to images in Figures S7, S9, S11, and S16.

Reviewer #3 (Remarks to the Author):

The work describes the formation of Möbius strips by self-assembly of chiral compounds/mesogens. The work is chemically sound. The results show the down-to-up chirality transfer by self-assembly of an asymmetric centre (chiral compound) to a morphologically chiral object and are highly important. This synthetic performance is an important proof of concept. However, major revision is necessary before publication. Therefore, in my opinion the manuscript should be published. However, several issues should be presented and discussed in a revised version. Specifically:

The following additional information must be included in the revised version

a) The topography of the ribbon: thickness is well discussed, length and curvatures too, but I could

not find the approximate value or range of its wideness.

The authors should take into account in the results discussion and according to the answer of the former two points (specially point (b)) the following comments.

a) It is not reasonable to expect that the chirality transfer from one asymmetric centre to a chiral object proceeds through structurally low defined aggregates such as those represented in Fig. 3. The planar layer assembly correspond in fact to a well define structure that, when X-ray diffraction at low distances could be performed, should show a specific chiral space point group (one of the 65 Söhncke groups). The building blocks would be the HA- of Fig. S13. This molecular structures should show frozen conformations of the aromatic and azo groups leading to additional interactions. Packing of these diastereoisomers will conform the unit cell. The packing could originate axial chirality that would be transferred to the bend and curvature of the spirals.

b) The toroid despite the many twists does not lead to additional bends or knots. The internal architectures (space point group) is the concrete framework that determines the twist, the curvate and that the toroid does not collapse to additional bends as could be expected for the small thickness and high diameter of the toroid (case of nucleic acids)

c) I could not find any statement on the presence or absence of toroid's showing zero (surely not present except for the case of the racemic toroid (Fig 2i)) or a number pair of twists. If these non-Möbius spiral toroids are not present, an important point is missing in the discussion on the down-to-up chiral transfer mechanisms. The absence would indicate an axial or planar chirality that needs of an impar number of twists to reproduce a continuous pattern in the translation/glide of the space point group.

c) The ribbon surface probably is composed by the polar groups and would show the same composition at both surfaces.

Less important.

The results on racemic mixture of compounds point to that system do not form racemic conglomerates but racemic crystals when the chiral building blocks are present as racemate

Abstract. Line 25

The L and D molecules are not "controlling" themselves are part of the mesogen. "Controllling" recalls the role of a chiral dopant.

Line 41

"violate" is not the right definition. The rule is "inverted" both for aromatic and antiaromatic is a consequence that topology change.

Figures EM. The number in the μm scale bar are missing in several figures

Response to reviewer comments point-to-point

Reviewer #1 (Remarks to the Author):

1. This paper describes the formation of cyclic nanofibers by glutamate amphiphiles functionalized with azobenzene. In particular, the authors focus on the twisting of the circular structures and claim that they are Moebius strips on the assumption that they twist an odd number of times. The formation of the microscale circular structure is interesting (but not new, as reported for polymers, small molecules, biomolecules, and carbon nanotubes), but the main point of this paper, especially the identification of the Moebius structure, remains a major question. Although the authors claim generality of the present example many times, the given example is a highly specific self-assembly phenomenon, and the discussion is not in depth with many ambiguous points in principle (given below). The authors presented good figures, carried out complex calculation and intricate analysis, but failed to show the principal mechanism. Therefore, the referee does not agree the publication of this paper in Nature Communications. In my opinion, Langmuir of ACS is the best journal to submit this paper.

Answer 1: We thank the referee for his/her comments very much, which is very helpful to further improve our manuscript. Inspired by your comments, we have done additional experiments and proposed a more reasonable self-assembly mechanism for the formation of the Moebius strips and nano-toroidal fibers with controlled helicity.

Just as you pointed out, the formation of a microscale circular structure is interesting (a recent example, S. Yagai et al. Self-assembled poly-catenanes from supramolecular toroidal building blocks. *Nature*, **2020**, 583, 400-405), not only for satisfying curiosity of people but also for understanding myriad chiral toroidal structures in nature such as circular DNA and proteins, which has attracted a lot of attentions among chemistry and biology communities. However, different from molecular circular structures such as crown ether, which are mainly formed by intramolecular covalent cyclization via end-to-end collision (usually in the presence of a template), the formation of larger-size self-assembled circular structures (nm to μm scale) is much more challenging: first, a self-assembled segment with proper curvature should be formed before cyclization, for which we still lack accessible methods; second, the cyclization process is hard to control due to the absence of a suitable template; third, the large size of self-assembled toroid structures increases the possibility of ring collapse especially if the torsion stems from chiral twist. Although several graceful self-assembled circular structures were successfully constructed, however, toroidal structures with chirality characteristics and helicity control have been rarely reported. As special topological toroids, Möbius strips are much harder to prepare due to the introduction of twist torsion. To the best of our knowledge, self-assembled Möbius strips with well-controlled helicity are not reported to date.

As a continuation of our ongoing endeavor in developing novel self-assembled chiral structures and their functions, in this manuscript, we utilized a supramolecular

cyclization method to obtain chiral nano-toroids and Möbius strips from bended twisted nanofibers, which were self-assembled from chiral glutamic amphiphile molecules. This method was based on the unique properties of glutamic amphiphiles: first, glutamic amphiphiles functionalized with aromatic moieties can easily form chiral twisted structures (*Adv. Mater.* 2016, 28 (6), 1044-1059., *Chem. Commun.* 2018, 54, 4513-4516.); second, the curvature of these twisted nanostructures can be changed due to the pH-responsive ability of glutamic head (*Chem. Eur. J.*, **2014**, 20,15419-15425); third, the helical chirality of these twisted structures can be effectively visualized through electron microscopy techniques and further supported by circular dichroism spectroscopy. Therefore, it should be possible to form and recognize chiral toroidal structures and Möbius strips via loop closure of bended nano-twisted fibers. The ratio of chiral toroidal structures among the overall nanofibers can be improved through both structural engineering of glutamate molecules and optimization of self-assembly protocols.

As for the identification of chiral circular nanofibers and Möbius strips, we have applied better marks to help readers to recognize these novel structures. Inspired by your comments, we have carried out spectroscopic study for self-assembly process and statistical analysis of chiral circular structures observed from electron microscopy techniques. Through a heating-cooling process, these amphiphile compounds are capable of forming twisted nanofibers under the assistance of multiple non-covalent interactions. It was found that the change of pH value drives the bending of these twisted fibers by cooperative adjustment of azobenzene stacking and amide hydrogen bonding. Further lowering amphiphile concentration efficiently shortened the length of bended fibers, which promoted the intrafiber end-to-end cyclization, leading to the formation of chiral toroidal nanofibers in which Möbius strips could be directly visualized by SEM technique.

Overall, in the present work, we have obtained larger-sized Möbius strips and nanotoroids through self-assembly and clearly visualized these nanostructures using SEM and AFM techniques. Second, we have controlled the helicity of Möbius strips for the first time by using the starting building block with opposite molecular chirality. Third, based on the new experimental results as inspired by your comments, we proposed a more reasonable self-assembly and cyclization mechanism in our revised manuscript, which include the efficient packing of the amphiphiles in a curved way and the pH triggered end-to-end cyclization.

2. Moebius structures might be formed in many linear fibers and the authors mention the term “yield” many times in the manuscript, but they have not quantitatively analyzed the population of the structures.

Answer 2: We agree with you that the term “yield” is not used properly. To evaluate the population of chiral toroidal structures among the whole nanofibers, we statistically analyzed the morphological parameters of toroidal fibers. As shown in revised Fig. S8, the ratio of circumference of chiral toroidal structures to the length of whole nanofibers including toroids and uncyclized fibers is calculated to be about 50 %, which is obviously larger than that of L-3 assemblies at higher concentration

(Fig. S5). We further statistically analyzed the distribution of twist number, diameter and fiber width among 95 nanotoroids observed through scanning electron microscopy images of L-3 amphiphile. As shown in revised Fig. S9-11, when the twist numbers were small (≤ 3), we can clearly recognize the exact twist numbers and helicity of these chiral toroids. Singly twisted toroids are the dominant species among those with small twist numbers as shown in revised Fig. 2i and S9 (supporting information). Diameter and fiber width analysis indicated that the diameters of majority toroids are among 500-2500 nm and fiber width are among 60-230 nm. According to the diameters, the average circumference of these circular fibers is among 1.6-7.8 μm , which is obviously shorter than the length of long fibers at high concentration (Fig. S5). These results implied that the length of bended fibers has significant influence on successful cyclization, therefore, the formation of chiral circular structures can be promoted by lowering the amphiphile concentration to reduce the amount and shorten the length of bended fibers.

3. The authors describe the preparation of the Moebius structure as a simple strategy. But, which section of the preparation is strategic? What are the motivations of such molecular design and preparation protocol to obtain Moebius structure? Do the circular structures form accidentally? Which factor drives the cyclization of nanofibers?

Answer 3: Thank you very much for the question. We have long been working on the self-assembly of the pi-conjugated chiral amphiphiles and accumulated much experience in the molecular design for chiral self-assembly. Here, the designed molecules contain three main parts, an alkyl tail, a planar aromatic azobenzene and a chiral glutamic acid head. The motivations of such molecular design are as following:

1. Though changing the length of alkyl chain, the hydrophobic/hydrophilic balance of the amphiphiles can be effectively adjusted, which significantly influences their self-assembly capability and resultant nanostructures in water.

2. The glutamic acid head can provide chiral information and facilitate a series of chiral assemblies. Most importantly, the ionization process of glutamic acid is varied at different pH values and influences the molecular length and components of the amphiphile as revealed by DFT calculations, which drives the bending and further cyclization of nanofibers through cooperative adjustment of multiple non-covalent interactions among amphiphiles.

3. The azobenzene moiety can promote the self-assembly process by intermolecular pi-pi interactions. Besides, the introduction of an aromatic moiety contributes to analyze self-assembly process through spectroscopic techniques, such as UV-vis absorption and circular dichroism spectra.

Through the integration of the three parts in a single molecule, chiral self-assemblies (twisted fiber) of glutamic amphiphile can be obtained. As we pointed out in answer 1, the two key requirements for the formation of chiral toroids and Moebius strips are the bending and cyclization of twisted fibers. The bending of twisted fibers can be realized by adjusting the pH value. The mechanism is that the proton disassociation process of glutamic acid head can be tuned by the change of pH

values, which will then significantly change the molecular configuration and length of amphiphile as supported by DFT calculation. At lower pH value, the glutamic acid head mainly exists in the form of carboxy acid groups (H_2A), whose molecular length is shorter than the ionized counterparts HA^- and A^{2-} . Especially at $pH \sim 2$, the major components of glutamic amphiphile solution are HA^- and H_2A according to analysis of Henderson-Hasselbalch equation ($pH = pK_a + \log([base]/[acid])$). The existence of shorter H_2A and longer HA^- might lead to the adjustment of assemblies, which provides the driving force for bending of bilayer structures and nanofibers.

Due to the absence of suitable template, the cyclization of bended fibers is challenging. Theoretically, the two terminals of a gradually elongating bended fiber will meet if the curved precursor is in a plane. However, the assemblies are growing in three-dimensional aqueous media, their elongation direction is generally random. A feasible method is lowering the concentration of amphiphiles, shortening the length and reducing the amount of twisted fibers, which will enhance the probability of intrafiber end-to-end cyclization by suppressing interfiber fusion.

Through the combination of these two simple methods, lowering pH values and reducing amphiphile concentration, the number of chiral toroids and Möbius strips are effectively improved (revised Fig. S8). Repetition tests demonstrated that the chiral toroids and Möbius strips could be found at various batches of amphiphile assemblies, showing that Möbius strip formation in this system was reliable and not accidental.

Please also see answer 1 for referee 2.

4. The identification of the Möbius structure must be difficult for readers because the ribbon-like structures are ambiguous, and therefore the definition of their face is also ambiguous. For example, are the structures shown in Fig. 1 g, 1k Möbius?

Answer 4: We apologize that the definition of their face of Möbius structures in our previous Figures are not clear, which made it difficult to identify. The structures shown in previous Fig. 1 g, 1k (Fig 2c, g in revised manuscript) cannot be assigned as Möbius strips due to tightly packed twists on the nanoring fiber, but they are chiral nanotoroids. In our revised Fig. 3, we have used a more easier fashion to elucidate the face direction of singly twisted Möbius strip. For detailed identification description, please see page 8, second paragraph.

5. Thickness of the fibers forming Möbius structures is ambiguous. Although it is clear from XRD that the internal structure is a bilayer motif, the periodicity of about 4 nm means a multilamellar structure. Therefore, the results of MD simulation with the single bilayer structure are not reasonable.

Answer 5: The thickness of nanotoroid fibers can be identified from AFM height profile, as shown in Fig. 2d and S14. The average thickness of fibers is about 110 nm. We also measured the average diameter (R) and width (w) of nanotoroid fibers by statistically analyzing 95 chiral nanotoroids, please see answer 2. These results indicated the average R/w ratio is among 8.3-10.0, which is consistent with the theoretical value obtained from MD simulation. We understand your doubt about the current MD simulation with the single bilayer structure. However, due to the

restriction of current computation capability, the 45*10*2 array (900 molecules) for simulation is already very time-consuming. Therefore, based on the agreement of theoretical and averaged experimental R/w values, we believe that the current MD simulation can provide useful information for understanding self-assembly and bending processes of twisted fibers.

6. The essential mechanism for the odd-number times of twisting is not described.

Answer 6: We have revised our manuscript by analyzing the proportion of nanotoroids with different twist numbers, including odd- and even-number times of twisting. Experimental data demonstrated that the singly twisted Moebius strips were the dominant species among those with small twist numbers as shown in Fig. 2i and S9 (supporting information). Based on the newly obtained spectroscopic and statistical data, a possible self-assembly and cyclization mechanism has been proposed in our revised manuscript, page 12, first paragraph and page 15, first paragraph.

7. The authors analyze the specific chirality of Moebius strips using CD spectroscopy. However, this analysis is questionable because the CD spectra contain large contribution of uncyclized fibers. Therefore, the CD-spectra cannot explain the specific chiral contribution of the Moebius structures.

Answer 7: The amphiphile self-assemblies were composed of chiral toroids and uncyclized fibers. Their helical chirality can be clearly recognized by SEM and AFM images, which is a common method for identification of supramolecular chirality. Results indicated that both chiral toroids and uncyclized fibers showed the same helicity. Therefore, the overall chiral signals revealed by CD spectra could be used to identify the chiral contribution of both chiral toroids (including chiral Moebius structures) and uncyclized fibers.

8. There are many additional errors in the manuscript, for example,

P5, (Figs. 1g, 1h, 1j, 1l, S3a) is (Figs. 1g, 1k, 1j, 1l, S3a)?

P7, (Fig. S5) is (Fig. S4)?

P7, (Fig. S6) is (Fig. S4)?

Answer 8: Thank you for careful review, we have corrected corresponding errors in our revised manuscript.

Reviewer #2 (Remarks to the Author):

In my personal opinion, publication of images of self-assembled Moebius strips (and their helicity control) themselves is valuable enough. This is professional work on masterpiece products of self-assembly. I respect this work. I basically recommend publication of this work in Nat. Commun. This work is well done and well considered. However, some revisions are necessary.

Answer: We thank the referee for his/her strong support for publication of our work.

1. The contents of this work probably satisfy curiosity of general readers. However, it might not be perfectly satisfactory for specialists in field of molecular assembly. Molecular-structure-based explanation to rationalize molecular interaction in structure-level to mesoscopic morphologies is not so clear. At least, such molecular-interaction-based explanation with chemical formulae had better be presented (proposed) in supporting information.

Answer 1: Thank you for the suggestion. We have measured UV-vis, CD, XRD and FT-IR spectra of amphiphile self-assemblies at different pH values to obtain more information for assembly process. UV-vis spectra showed at the hypsochromic shift of azobenzene moiety during self-assembly process is more obvious (from 346 to 322 nm, Fig. 5b) when lowering pH values from 6 to 2, which implied a much more compact aggregation at lower pH values. The compact aggregation of azobenzene would suppress its photo-isomerization ability, this was well-proved by the azobenzene photo-isomerization experiments. As shown in revised Fig. 5f, the CD spectra of L-3 assemblies at pH 6 showed photo-responsiveness under UV 365 nm irradiation, while it shows no obvious photo response when pH value was lowered to 2. These CD results indicated that the π - π stacking is stronger at lower pH, which is consistent with absorption spectra.

CD spectra of L-3 assemblies at pH = 6 showed a normal negative Cotton effect (Fig. 5f, black dotted line), while a bisignate Cotton effect was observed for L-3 assemblies at pH = 2 (Fig. 5e). According to exciton chiral theory (Harada, N. & Nakanishi, K. Circular dichroic spectroscopy: exciton coupling in organic stereochemistry, University science Books, Mill Valley, 1983), the negative bisignate Cotton effect showed in Fig 5e implied a counterclockwise screw sense of the two transition moments of adjacent azobenzene moieties. Therefore, a reasonable aromatic stacking mechanism based on these data was proposed (Fig. 7a-b, S16). At higher pH value, the bilayer structures adopt a parallel stacking mode and further assemble into twisted multi-bilayers and fibers through hierarchical self-assembly (Fig. 7a). At lower pH values, the major components of amphiphile molecules are H₂A and HA⁻, the shorter molecular length of H₂A led to a rotation between two adjacent bilayers (Fig. 7b, S16). As a result, the CD signal showed an obvious bisignate Cotton effect with a crossover at the absorption maximum region of azobenzene. Besides, the rotation of bilayer stacking should also influence the hydrogen bonding of amide groups. This was well supported by FT-IR spectra, which showed that the C=O stretching vibration of amide bond was changed from 1621 to 1624 nm when lowering pH values from 6 to 2, indicating that the hydrogen bonding between amide

groups was weakened (Fig. 5d). We have added detailed discussion in our revised manuscript and supporting information.

2. Similarly, description in Introduction had better include more about deep molecular assembly science with citing recent comprehensive papers on self-assembly (see, Bull. Chem. Soc. Jpn. 91, 623-668 (2018), Mol. Sys. Des. Eng. 4, 11-28 (2019).). Current Introduction is rather too much oriented topology and appearance on Moebius strips.

Answer 2: We have added several recent review papers (ref. 27-29) on self-assembly and revised the introduction part by adding more description on chiral toroid structures and self-assembly.

3. Please avoid several easy mistakes. References 35-37 remains at unnatural place. Reference styles are not perfectly unified (comma, colon, etc)

Answer 3: Thank you for careful review, we have corrected these mistakes in our revised manuscript.

4. Please remove unnecessary grey background from Figure 2j. I do not care about imperfect nature of CD spectra of chiral assemblies. It may happen.

Answer 4: We have moved the new CD spectra to Fig. 5e, and added the CD spectra of L-3 assemblies at pH = 6 (also including CD spectra change under UV and visible irradiation).

5. Please provide clear scale bars and scale values to images in Figures S7, S9, S11, and S16.

Answer 5: We have added scale bars and values for all microscopic images in our revised manuscript and supporting information.

Reviewer #3 (Remarks to the Author):

The work describes the formation of Möbius strips by self-assembly of chiral compounds/mesogens. The work is chemically sound. The results show the down-to-up chirality transfer by self-assembly of an asymmetric centre (chiral compound) to a morphologically chiral object and are highly important. This synthetic performance is an important proof of concept. However, major revision is necessary before publication. Therefore, in my opinion the manuscript should be published. However, several issues should be presented and discussed in a revised version. Specifically:

The following additional information must be included in the revised version.

1. The topography of the ribbon: thickness is well discussed, length and curvatures too, but I could not find the approximate value or range of its wideness.

Answer 1: Thank you for support to our work. We have statistically analyzed the average diameter and width of 95 chiral circular structures. For details, please see revised manuscript, page 5, Fig. 2j-k.

The authors should take into account in the results discussion and according to the answer of the former two points (specially point (b)) the following comments.

2. It is not reasonable to expect that the chirality transfer from one asymmetric centre to a chiral object proceeds through structurally low defined aggregates such as those represented in Fig. 3. The planar layer assembly correspond in fact to a well define structure that, when X-ray diffraction at low distances could be performed, should show a specific chiral space point group (one of the 65 Söhncke groups). The building blocks would be the HA⁻ of Fig. S13. This molecular structures should show frozen conformations of the aromatic and azo groups leading to additional interactions. Packing of these diastereoisomers will conform the unit cell. The packing could originate axial chirality that would be transferred to the bend and curvature of the spirals.

Answer 2: Thank you for your insights, which inspired us for more detailed mechanism study on chirality transfer. We first investigated the CD spectrum of L-3 amphiphile in THF solution, no obvious CD signal could be observed at azobenzene absorption region. This result proved that chirality transfer cannot be realized at the molecular state. For self-assemblies, the major components of amphiphile molecules are H₂A and HA⁻ at lower pH values, the shorter molecular length of H₂A led to a rotation between two adjacent bilayers as supported by the CD signal showing a bisignate Cotton effect with a crossover at the absorption maximum region of azobenzene. However, at higher pH value, the tiny molecular length difference of HA⁻ and A²⁻ lead to the formation of straight twisted fibers through hierarchical self-assembly, which gave a normal negative CD signal. The CD spectra showed that the relative orientation of azobenzene groups at different pH values varied significantly, which played a leading role in chirality transfer from molecular level to supramolecular level. But unfortunately, due to the amphiphilic nature of these molecules, their space point group information (which usually suitable for single

crystals) was not obtained by X-ray diffraction technique. For detailed discussion on self-assembly and chirality transfer mechanism, please see revised manuscript, page 11, last paragraph and Fig. 7.

3. The toroid despite the many twists does not lead to additional bends or knots. The internal architectures (space point group) is the concrete framework that determines the twist, the curvate and that the toroid does not collapse to additional bends as could be expected for the small thickness and high diameter of the toroid (case of nucleic acids)

Answer 3: Just as you pointed out, the small thickness and high diameter of the toroid should lead to collapse, which is supported by the fact that all the observed toroidal fibers with diameter less than 4 μm , the average diameter of majority chiral toroids is between 500-2500 nm. According to the analysis of spectroscopic data, the cooperation of multiple non-covalent interactions such as pi-pi stacking between azobenzene moiety, H-bonding between amide groups and Van der Waals forces between alkyl chains is responsible for the stabilization of toroidal fibers. Please also see answer 1 to referee 2.

4. I could not find any statement on the presence or absence of toroid's showing zero (surely not present except for the case of the racemic toroid (Fig 2i)) or a number pair of twists. If these non-Möbius spiral toroids are not present, an important point is missing in the discussion on the down-to-up chiral transfer mechanisms. The absence would indicate an axial or planar chirality that needs of an impar number of twists to reproduce a continuous pattern in the translation/glide of the space point group.

Answer 4: In the previous version, we put representative circular structures in the supporting information. And in the manuscript, we focused on chiral Mobius strips. Obviously, this caused some misunderstanding. Actually, circular fibers with both odd and even numbers of twists could be observed (revised Fig. S9-11).

5. The ribbon surface probably is composed by the polar groups and would show the same composition at both surfaces.

Answer 5: We agree with your comment. The surface of most amphiphile assemblies is composed of hydrophilic groups in aqueous media. And the pH-sensitivity of the assemblies also indicates the glutamic head are in the ribbon surface. For MD simulation, due to the restriction of current computation capability, we simplified the pre-assembled bilayer. Please also see answer 5 for referee 1.

Less important.

6. The results on racemic mixture of compounds point to that system do not form racemic conglomerates but racemic crystals when the chiral building blocks are present as racemate

Answer 6: From our experimental results, the racemate didn't show chiral self-sorting. Instead, achiral fibers and nanotoroids were formed for L3/D3 mixture (molar ratio 1:1) as supported by microscopic images and CD spectra.

7. Abstract. Line 25

The L and D molecules are not "controlling" themselves are part of the mesogen. "Controlling" recalls the role of a chiral dopant.

Answer 7: We have changed this sentence to "the helicity of the Möbius strips and nano-toroids stems from the molecular chirality of glutamate molecules. Therefore, M- and P-helicity Möbius strips could be formed from L- and D-amphiphiles, respectively".

8. Line 41

"violate" is not the right definition. The rule is "inverted" both for aromatic and antiaromatic is a consequence that topology change.

Answer 8: Thank you for your comments. We have rewritten the introduction part in our revised manuscript.

9. Figures EM. The number in the μm scale bar are missing in several figures.

Answer 9: We have added scale bars and values for all microscopic images in our revised manuscript and supporting information.

REVIEWERS' COMMENTS

Reviewer #1 (Remarks to the Author):

I have read the revised version of manuscript. The authors have added a number of new data, which strongly supports the claims made by the authors. Also I am satisfied with the reply to my previous comments. If the other two reviewers are also satisfied with the author's rebuttal to their comments (seem to be more critical in comparison with mine), I would like to recommend the publication of this work in Nature Communication with the revised version.

Reviewer #2 (Remarks to the Author):

I found significant efforts by the authors to revise this manuscript. Revisions are sufficient enough and actually appropriate. The revised version becomes acceptable.

Reviewer #3 (Remarks to the Author):

The thoroughly revised manuscript improves the first work report. The inclusion and new discussion of more experimental details gives a better insight in the formation of cyclic strips and in the reasons for the formation of Möbius surfaces.

All my previous comments have been answered or reflected in the revised text, therefore the present brief report.

In my opinion, the authors could be more decided in the discussion of self-assembly model when the formation of strips with two distinct surfaces can only cycle through the formation of Möbius surfaces. However, this belongs to the author decision.

In the basis of the scientific merits of the submitted work my recommendation for acceptance in Nature Communications.

Response to reviewer comments point-to-point

Reviewer #1 (Remarks to the Author):

I have read the revised version of manuscript. The authors have added a number of new data, which strongly supports the claims made by the authors. Also I am satisfied with the reply to my previous comments. If the other two reviewers are also satisfied with the author's rebuttal to their comments (seem to be more critical in comparison with mine), I would like to recommend the publication of this work in Nature Communication with the revised version.

Answer: Thank you very much for your support for publication of our work.

Reviewer #2 (Remarks to the Author):

I found significant efforts by the authors to revise this manuscript. Revisions are sufficient enough and actually appropriate. The revised version becomes acceptable.

Answer: We thank the reviewer for his/her support for publication of this work.

Reviewer #3 (Remarks to the Author):

The thoroughly revised manuscript improves the first work report. The inclusion and new discussion of more experimental details gives a better insight in the formation of cyclic strips and in the reasons for the formation of Möbius surfaces.

All my previous comments have been answered or reflected in the revised text, therefore the present brief report.

In my opinion, the authors could be more decided in the discussion of self-assembly model when the formation of strips with two distinct surfaces can only cycle through the formation of Möbius surfaces. However, this belongs to the author decision.

In the basis of the scientific merits of the submitted work my recommendation for acceptance in Nature Communications.

Answer: We are grateful to the reviewer for his/her recommendation for acceptance.